# Fungal Biocontrol Agents in the Management of Postharvest Losses of Fresh Produce—A Comprehensive Review

**DOI:** 10.3390/jof11010082

**Published:** 2025-01-20

**Authors:** Phathutshedzo Ramudingana, Ndivhuho Makhado, Casper Nyaradzai Kamutando, Mapitsi Silvester Thantsha, Tshifhiwa Paris Mamphogoro

**Affiliations:** 1Gastro-Intestinal Microbiology and Biotechnology Unit, Agricultural Research Council-Animal Production, Private Bag X02, Irene, Pretoria 0062, South Africa; ramudinganap@arc.agric.za; 2Department of Microbiological Pathology, Tuberculosis Research Unit, Sefako Makgatho Health Sciences University, Molotlegi Road, Ga-Rankuwa, Pretoria 0204, South Africa; nmakhado@yahoo.com; 3National Health Laboratory Services, Dr George Mukhari Tertiary Laboratory, Pretoria 0204, South Africa; 4Department of Plant Production Sciences and Technologies, University of Zimbabwe, P.O. Box MP167, Mount Pleasant, Harare 0263, Zimbabwe; kamutandocn@gmail.com; 5Department of Biochemistry, Genetics and Microbiology, University of Pretoria, Private Bag X20, Hatfield, Pretoria 0028, South Africa; mapitsi.thantsha@up.ac.za

**Keywords:** biological control agents, postharvest loss, fungal antagonist, fresh produce

## Abstract

Postharvest decay of vegetables and fruits presents a significant threat confronting sustainable food production worldwide, and in the recent times, applying synthetic fungicides has become the most popular technique of managing postharvest losses. However, there are concerns and reported proofs of hazardous impacts on consumers’ health and the environment, traceable to the application of chemical treatments as preservatives on fresh produce. Physical methods, on the other hand, cause damage to fresh produce, exposing it to even more infections. Therefore, healthier and more environmentally friendly alternatives to existing methods for managing postharvest decays of fresh produce should be advocated. There is increasing consensus that utilization of biological control agents (BCAs), mainly fungi, represents a more sustainable and effective strategy for controlling postharvest losses compared to physical and chemical treatments. Secretion of antifungal compounds, parasitism, as well as competition for nutrients and space are the most common antagonistic mechanisms employed by these BCAs. This article provides an overview of (i) the methods currently used for management of postharvest diseases of fresh produce, highlighting their limitations, and (ii) the use of biocontrol agents as an alternative strategy for control of such diseases, with emphasis on fungal antagonists, their mode of action, and, more importantly, their advantages when compared to other methods commonly used. We therefore hypothesize that the use of fungal antagonists for prevention of postharvest loss of fresh produce is more effective compared to physical and chemical methods. Finally, particular attention is given to the gaps observed in establishing beneficial microbes as BCAs and factors that hamper their development, particularly in terms of shelf life, efficacy, commercialization, and legislation procedures.

## 1. Introduction

It is expected that population growth in Africa will reach 2.55 billion by the year 2050 [1]. However, in the previous 20 years, production of food in Africa has lagged behind population growth due to low productivity and high postharvest losses of farm produce [1]. Increasing the production of food has therefore become a priority in developing countries. Additionally, a reduction in postharvest losses is also considered as a viable solution in increasing food availability at household level. Increasing food availability neither requires additional resources nor places additional burden on the environment [2]. Different methods, which include physical, biological, and chemical treatments, are in place for the management and control of losses incurred during the harvesting of agricultural products; however, the measures in place have not helped in boosting food security in Africa [3].

The following are some physical and chemical methods used to manage postharvest losses: (i) temperature and humidity management, where cold chain management uses optimal temperature and relative humidity throughout the postharvest production pathway [4]; (ii) intermittent warming, where for example, mature, green and early breaker fruits are treated to intermittent warming for one day per week at 20 °C and then stored at 6, 9, and 12 °C for 28 days to reduce microbial decay [5]; (iii) modified/controlled atmosphere [6], which prevents storage rot through the use of films and coatings to alter the air composition around the produce and/or to reduce water loss [7]; and (iv) UV treatments [8], which inactivate the microorganisms and prevent microbial replication through creation of pyrimidine dimers in the DNA [7]. The main disadvantages of physical methods are that the produce could be damaged during treatment, thereby exposing the crop produce to infection, and furthermore, they are relatively expensive and thus may not be afforded by smallholder farmers [9].

Chemical treatments used for the control and management of diseases that cause spoilage of fresh produce include the use of fungicides and antibiotics. Fungicides and antibiotics are applied on the surface tissues of produce using smart fog and ultrapure systems, as well as biocoating, for controlling infections already established [7]. They serve to inhibit pathogen proliferation by safeguarding plant produce against infections that primarily arise during handling and storage [10]. Postharvest fungicides are applied as dips [11], sprays [11], fumigants [12], coatings [13], treated box liners, or in waxes and wraps [14]. Amongst others, fungicides that are commonly applied include triazoles (e.g., imazalil and prochloraz) and benzimidazoles (e.g., benomyl and thiabendazole) [15].

Smart Fog [16] and Ultrapure systems [17] are dry fogging systems used to introduce antimicrobial sprays onto fruit surfaces. The major challenge with the use of these systems is that the bottom of the fruit surfaces, which will be in contact with the container, are often unexposed to the environment and remain unsterilized. Biocoating is the application of edible food coating (e.g., oils) to uphold the standard of the produce [7]. These edible coats retard ethylene emission and enhance texture quality thereby enhancing the shelf life of the plant produce [7,18].

Even though, to date, chemical treatments are considered the primary method used to control the postharvest decay of fresh produce, their main limitation is that some chemicals lack approval for postharvest treatment [19]. As a result, these chemicals are withdrawn from the market due to apprehensions over their potential toxicological risks [20]. Additionally, there are serious public concerns over the use of fungicides and antibiotics, the impact of their ecotoxicity and human toxicity, their associated exorbitant expenses and development of antibiotic resistance in pathogens. Furthermore, the use of chemicals on fresh produce necessitates intricate utilization, since their efficacy is influenced by both abiotic and biotic factors, and demands advanced industrial manufacturing. These issues prompted researchers to explore alternative approaches [3,20,21]. The ideal alternative approaches should therefore be safe for both the environment and consumers [20]. Biological control agents are regarded as a potential management option with these desirable features.

Biocontrol is the use of naturally occurring microorganisms or controlled microflora and (or) their antimicrobial products to enhance the safety of fresh produce while extending its shelf life [22]. Examples of biocontrol agents include antagonistic microorganisms and animal-derived and natural plant compounds [23]. Amongst fungi, yeasts and yeast-like organisms with antagonistic properties isolated from the surface of leaves, vegetables, and fruits are used as biological control agents (BCAs) [3,20,24]. The application of fungal BCAs ensures the production of high-quality fresh produce and prolonged shelf life [25]. The drawbacks of chemical applications render antagonistic fungi as the preferred strategy for biocontrol [26]. Furthermore, these antagonists may originate from non-target, non-toxic microbes, rendering them readily acceptable to ‘eco-friendly’ customers [20]. This is a novel, cost-efficient strategy that farmers should be urged to implement as an alternative to chemical treatments. The effectiveness of biocontrol agents at low concentrations reduces their ecological footprint, which indicates that their use does not place a high demand on resources, thus making them a more sustainable practice; for example, a study by Ansari et al. [27] demonstrated that the biocontrol agent *T. harzianum* was effective at low concentrations in controlling *Fusarium oxysporum* f. sp. *lycopersici* (FOL), underlining the potential of *T. harzianum* biocontrol as a cost-effective and environmentally sustainable alternatives for managing FOL [27,28].

Biocontrol agents operate through multiple modes of action, consequently diminishing the emergence of resistance. Additionally, they are accessible for local-scale producers due to their ease of use and are produced in controlled industrial settings [1,3]. Thus, it is anticipated that the application of biocontrol products will, in the near future, prevail over the use of agricultural chemicals. Some of advantages of commercial BACs are that they (i) can be stored for more than 12 months at ambient temperature and for more than 24 months at 4–6 °C; (ii) are applicable in organic farming; (iii) reduce the risk of appearance of resistant strains; (iv) are active at low temperatures; (v) are not phytotoxic; (vi) have residues that are not typically subject to standard restrictions or maximum residue limits (MRLs) that apply to other chemicals; (vii) do not affect vinification; (viii) can fit to any integrated pest management (IPM) program; (ix) are harmless to beneficials; (x) are safe for both humans and the environment [29].

The aim of this review is to compare the existing methods for postharvest management of fresh produce decay and to highlight evidence supporting fungal biocontrol agents as a viable alternative for control of postharvest diseases. We anticipate that this review will encourage acceptance and support for the introduction of fungal BCAs as a healthier, affordable, and more environmentally friendly alternative as a replacement for synthetic chemicals in the management of postharvest fresh produce decay.

## 2. Characteristics of Ideal Fungal BCAs

Several good character traits desired in fungal BCAs are documented [1,7]. An ideal fungal antagonist for management of postharvest decay must be (i) able to colonize and persist on the commodity at effective levels; (ii) compatible with other postharvest practices, processes, and chemicals; (iii) effective under adverse environmental conditions including low temperatures as well as, in some cases, controlled-atmosphere conditions; (iv) effective at low concentrations; (v) genetically stable; (vi) amenable to large-scale production using low-cost substrates; (vii) easy to dispense; (viii) appropriate for a formulation that maintains stability over a lengthy period of time (have an extended shelf life); (ix) non-fastidious in its nutrient requirement; (x) unable to produce metabolites deleterious to human health; (xi) not harmful to the host commodity; (xii) compatible to commercial processing procedures; (xiii) resistant to pesticides; (xiv) adequately effective over a broad variety of vegetables and fruits; and (xv) adequately effective at combating a wide range of pathogens [1,3,30]. Based on the traits of an ideal BCA listed above, species of *Trichoderma*, mainly *T. harzianum*, continue to hold a significant position among BCAs in a wide range of crop and disease management strategies. Whether used as a standalone treatment or in combination with other ingredients to form a comprehensive IPM strategy, they are commonly applied as soil treatment to control soil-borne pathogens such as *Fusarium* and *Pythium*, *Phytophthora*, and *Rhizoctonia* [31]. Hence, more studies have focused on the isolation, identification, and characterization of fungi as potential BCAs in managing postharvest losses of fresh produce [32,33,34].

## 3. Factors Influencing Fungal BCAs Effectiveness

### 3.1. Microbial Inoculum Pressure

It has been reported that pathogen concentrations determine the sensitivity and efficacy of the BCAs [1], where an increase in the pathogen spore concentration results in a decrease in the efficacy of the BCAs [35]. Biological control agents should have the ability to adhere to plant tissues and pathogens and achieve a certain cell concentration (spores and hyphae) at the infection site [1,30]. Ideally, the concentration of the BCA should be higher than that of the target pathogen. For example, *Fusarium oxysporum* strain Fo47 was reported to be effective as an antagonist only when introduced at concentrations 10–100 times higher than that of the pathogen [36]. Reportedly, *Fusarium oxysporum* CS-20 significantly reduces wilt incidents in tomato at concentrations of up to 1000 times higher than the concentration of the pathogen [36]. Therefore, minimizing exposure of fresh produce to pathogens is essential for the success of postharvest biocontrol [37].

### 3.2. Formulations of Fungal BCAs

Commercial prepared BCAs should retain their efficacy and possess adequate shelf life in their final formulated form. Fungal biocontrol product preparations are available in both dry and liquid formulations [38]. Both types of formulations should include the exogenous protectant to improve the efficacy of ecological strain in biocontrol and improve cell viability [39]. The two formulations and studies conducted on them are discussed below.

#### 3.2.1. Dry Formulation

Freeze-drying is a widely employed technique for obtaining desiccated microorganism preparations [39,40]. A study by Lee et al. [41] assessed the efficacy, viability, and stability of freeze-dried *Candida sake* across various rehydration and protective media [41,42]. They reported that the highest biocontrol activity was attained when 10% skim milk or 10% lactose was used a protective agent. Furthermore, incorporating 1% peptone during the rehydration of cells, before their application, also protected cell viability. However, use of nutrient protectants at such elevated concentrations would not be viable from an economic standpoint during large-scale production. In comparison, cryopreservation of cells in liquid nitrogen caused the most severe damage to the cells, decreasing their viability by 10%. Freeze-drying is comparatively costly and necessitates the use of specialized machinery for batch drying. As a result, numerous studies have investigated alternative drying techniques for the production of biocontrol agents such as fluidized bed drying, vacuum drying, and spray drying (Table 1).

In their comparative analysis of various formulations methods, including vacuum drying, liquid formulations, and freeze-drying of *Pichia anomala* J121, Druvefors et al. [43] reported that freeze-drying of the yeast resulted in the highest viability. Another study investigated the efficacy of freeze-drying on *Rhodotorula glutinis* and *Cryptococcus laurentii,* using 5% or 10% of exogenous trehalose as a protective agent. Notably, the initial viability of the organisms reached 80% [43]. Trehalose concentrations, both exogenous and endogenous, were found to substantially enhance cell viability [44,45,46]. According to Lee et al. [41], the application of spray drying to *C. sake* led to a significant degree of cell injury, which resulted in significant decrease in cell viability.

Some of the advantages of a dry formulation include, among others, preservation of the product’s integrity through prevention of contamination during storage, extended durability during storage at room temperature, and ease of product storage and shipping [44,45].

#### 3.2.2. Liquid Formulation

Liquid media is frequently utilized as a method to generate biocontrol agents for commercial purposes (Table 1). The efficacy of liquid formulations has not been thoroughly assessed for fungal-based biocontrol products [39]. *Candida sake* CPA-1 prepared as a liquid formulation in sorbitol-modified medium, and subsequently stored in isotonic solution of trehalose was assessed for ability to preserve cell viability [47]. The results indicated that *C. sake* cells exhibited an approximate seven-month shelf life without significant viability or efficacy loss. The formulation of *Rhodotorula minuta* in a liquid formulation was assessed in an additional pilot-scale study whereby 10^9^ CFU/mL of the yeast was formulated in a phosphate buffer solution, to which xanthan (0.5%) and glycerol (20%) were added; such formulation effectively inhibited pathogen contamination [47,48]. Furthermore, an assessment was conducted on the viability of *P. anomala* in a liquid formulation supplemented with trehalose, glucose, or lactose [48]. Addition of either lactose or trehalose to the storage medium resulted in a significant degree of viability, sustained for 8–12 weeks across all evaluated temperatures (−20 °C–30 °C). In order to boost the survival and efficacy of *Pichia membranaefaciens* and *Cryptococcus laurentii* in a liquid formulation, sugar protectants (trehalose and galactose) were added together with L-ascorbic acid [49]. The durability of the yeast was prolonged to 90 days at 4 °C and 15 days at 25 °C.

Liquid formulations have several advantages, including low cost, since they do not require rehydration, addition of fillers, or drying. However, oxidative stress may cause decreased cell viability in liquid media during storage [48,49]. The antioxidant L-ascorbic acid can be used in liquid formulations to enhance the efficacy of sugar protectants [49].

**Table 1 jof-11-00082-t001:** Exemplary investigations on the composition of fungal BCA.

Formulation of BCAs	References
Dry	
Freeze-drying	
*Pichia anomala* J121	[43,50]
*Candida sake* CPA-1	[41]
*Rhodotorula glutinis*, *Cryptococcus laurentii*	[44,45]
Spray-drying	
*Candida sake* CPA-1	[46]
Fluidized bed-drying	
*Pichia anomala* J121	[50]
*Aureobasidium pullulans*	[51]
Vacuum-drying	
*Pichia anomala* J121	[43,50]
Liquid	
*Rhodotorula minuta*	[48]
*Pichia anomala* J121	[38,50]
*Candida sake* CPA-1	[47]
*Cryptococcus laurentii*, *Pichia membranaefaciens*	[49]

### 3.3. Delivery Systems of Fungal BCAs

Pathogens can infect fresh produce prior to harvest in the field; therefore, use of microbial antagonists could be beneficial in managing these diseases both at pre- and postharvest phases. It is therefore important to consider applying the antagonists prior to the harvest through soil drenches, seed treatment, irrigation systems, and foliar sprays, since they are able to establish a presence on the fruit surface as well, both in the agricultural field and throughout the storage process, and to remain on the fruit surface over time, thereby ensuring effective management of decay [51,52,53].

#### 3.3.1. Preharvest Application of Fungal BCAs

Since contamination by microbial agents on fresh produce occurs at any stage in food production, i.e., prior to harvest, during harvest and transportation, as well as in storage [54], it is vital to explore the best strategy for delivery of antagonistic fungi. Essentially, fungal antagonists are delivered either at preharvest or after harvest [3]. Preharvest application is usually conducted to facilitate pre-colonization of the fruit surface by the fungal antagonist, allowing for the antagonist to establish itself prior to the pathogen [52]. Hence, pathogen inhibition by a fungal antagonist is much more if the antagonist is administered before the occurrence of infection. Preharvest application helps control quiescent field infections, which are often difficult to control after harvest [3]. Even though some studies have reported that preharvest application may not be commercially feasible as a result of the low viability of antagonist under environmental conditions [3], numerous reports on its effectiveness have been documented [30,54].

A study by Huang et al. [55] reported that antagonist application of *Cryptococcus laurentii* and *R. glutinis* (Fresenius) reduced gray mold from 7% to nearly 1% and 13% to 4%, respectively, when applied for 3-week prior to harvest [55]. Kheireddine et al. [56] reported an approximately 50% reduction in the blue mold *Penicillium expansum* in wounded apples by *C. sake* CPA-1, inoculated two days prior harvesting, followed by storage of the apples in a refrigerated facility for a duration of four months [56].

Preharvest application of the antagonist *Aureobasidium pullulans* was reported to significantly reduce strawberry rot in storage [57], cherries [58], apples [59], and grapes [58]. Additionally, Zhang et al. [60] stated that preharvest application of *Pichia guilliermondii* reduced incidences of the green mold *Penicillium digitatum* on grapefruit [60]. Subject to field conditions, preharvest application of *Epicoccum nigrum* was shown to have an effect against citrus and peach brown rots [3]. Furthermore, Ayogu et al. [61] reported that application of antagonist *Metschnikowia fructicola* alone or in combination with sodium bicarbonate and ethanol, close to harvesting, significantly controlled postharvest decays of grapes [61]. Similarly, preharvest application of *M. fructicola* controlled pre- and postharvest rot in strawberry fruits [61]. Moreover, preharvest applications with various fungal antagonists such as *Trichoderma harzianum* [62], *E. nigrum* [63], and *Gliocladium roseum* Bainier [64] successfully controlled postharvest decay of strawberries where synthetic fungicides had proven to be ineffective.

#### 3.3.2. Postharvest Application of Fungal BCAs

Application of antagonistic microorganisms is common and appears to achieve efficacy in the management of postharvest diseases of fresh produce [3]. The antagonists are sprayed directly onto the surfaces of the produce or are applied by dipping and drenching [3]. This strategy has been proven to be an effective approach for strawberries [62], citrus [63], apples [65,66,67], peach [68,69], banana [70], mango [71], and tomato [72]. Nujthet et al. [73] reported that postharvest application of *Paecilomyces variotii* Bainier, *T. harzianum*, *Trichoderma viride* and *Gliocardium roseum* was more effective in controlling *Botrytis* and *Alternaria* rots than when applied at the preharvest stage [73]. Moreover, postharvest applications of *T. harzianum* was more effective in controlling *Aspergillus* and *Fusarium* rots in citrus fruits and potatoes. In yet other studies, significant reductions were reported for yeast antagonist in harvested fruits against storage decay caused by pathogens such as *P. italicum* and *P. digitatum* in citrus [37]; *B. cinerea* in apples [73]; *B. cinerea* and *P. expansum* in pears [74]; and *B. cinerea*, *Alternaria alternata* and *Rhizopus stolonifer* in tomatoes [35]. Postharvest applications of *Trichoderma viride* and *Debaryomyces hansenii* were effective against *P. digitatum* in citrus fruit [75,76].

Nevertheless, Ons et al. [3], reported that preharvest application has several drawbacks and does not effectively address disease management in fruit production for commercial purposes; for example, the effect of preharvest applications vary depending on the type of fruit, and the effect preharvest calcium application may reduce cell browning (CB) in some apple varieties, but not in others. Therefore, the use of postharvest fungal antagonists is a preferable and practicable technique for management of postharvest diseases of fresh produce [3].

## 4. Fungal BCAs in Postharvest Diseases of Farm Produce

Antagonistic microorganisms were successfully isolated, tested, and applied in biological management of postharvest infections in perishable crops against fungal spp., including *Candida*, *Cryptococcus*, and *Pichia* [3]. Others are currently still under varying degrees of development [3,77]. Despite an increase in the number of commercialized microbial biocontrol products on the market, these products still represent 1% of all the agricultural chemical sales [78]. Two available methods use fungi as antagonists to control and manage postharvest losses, including (i) application of antagonists already present on surfaces of the vegetables and fruits, e. g., natural fungal antagonists are naturally present on surfaces vegetables or fruits; following isolation, these fungal antagonists are employed for the purpose of postharvest decay management; (ii) the artificial introduction of postharvest pathogens’ antagonists [3,72].

The use of antagonistic yeasts to treat fresh produce is one of the most effective techniques for preventing postharvest diseases [30]. Various yeast species (Table 2), have been found to effectively control fruit decay. Nevertheless, the utilization of biocontrol alone does not yield economically satisfactory disease control in fruits [37,65,79,80].

The development of filamentous fungi as biocontrol agents for management of postharvest diseases is less advanced compared with yeasts. Yeasts generally outperform filamentous fungi in terms of rapid colonization, safety, environmental tolerance, versatility in mechanisms, adaptability, ease of use, and overall efficacy in biocontrol applications. Nevertheless, fungal antagonists, such as *Homoptera* and *Muscodor albus*, demonstrated a decrease in citrus fruit postharvest degradation [81,82,83].

**Table 2 jof-11-00082-t002:** List of fungal BCAs that are currently in use.

Antagonist	Disease	Target	Produce	Mechanism of Activity	Reference
*Cryptococcus laurentii* *C. albidus*	Gray mold Brown rot	*Monilinia fructicola Penicillium expansum Botrytis cinerea*	Apple Tomato Orange Sweet cherry Peach Strawberry	- Induce host defense responses - Inducing accumulation of resistance related enzymes - ROS tolerance - Attachment and lytic enzyme secretion	[84,85,86,87,88]
*Pichia guilliermondii* *P. menmbranaefaciencs* *P. guillermondii*	Gray mold Alternaria rot Rhizopus rots Blue mold Green mold Anthracnose	*Rhizopus stolonifer* *Botrytis cinerea* *Penicillium expansum* *Penicillium digitatum* *Collectrichum capsici* *Alternata alternata*	Tomato Apple Citrus Peach Grape Chillie	- Induce host defense responses - Attachment and lytic enzyme secretion	[3,25,37,72,84]
*Candida ciferii* (283) *C. sake* *C. saitoana* (240) *C. guilliermondii* *C. oleophila* (1–182)	Botrytis rot Penicillium rot Penicillium rot Gray mold	*Penicillium expansum* *Botrytis cinerea*	Apple Orange Tomato	- Induce host defense responses - Adjustment of population density	[89,90]
*Cystofilobasidium infirmominiatum*	Botrytis rot	*Botrytis cynerea*	Lemon	- Iron depletion - ROS tolerance	[91]
*Saccharomycess cereviciae* (N.826 and N.831)	Penicillium rot	*Penicillium expansum*	Grape	- Morphology change	[92]
*Metshnikowia fructicola* (NRRL Y-27328)	Botrytis rot	*Botrytis cynerea*	Grape	- Iron depletion	[61]
*Trichosporon pullulans*	Alternaria rot Gray rot	*Botrytis cynerea*	Cherry	- Production of lytic enzymes	[84]
*Pestalotiopsis neglecta*	Anthracnose	*Collectotrichum gloeosporoides*	Apricot	- Production of lytic enzymes	[93]
*Debaryomyces hansenii*	Rhizopus rot Alternaria rot Gray mold	*Botrytis cynerea* *Penicillium expansum*	Tomato	- Induction of host resistance - Competition for nutrients and space	[3]
*Rodotorula glutinis*	Blue mold Gray mold Alternaria rot Green mold	*Penicillium expansum Botrytis cinerea*	Apple Orange Pear Strawberry Sweet cheery	- Competition for nutrients and space - Site exclusion	[3,67,87,94,95,96]
*Trichordema harzianum*	Anthracnose Gray mold	*Collectrichum muse* *Collectotrichum gloeosporoides*	Banana Strawberry Pear Kiwi Grape	- Production of antibiotics	[97,98]
*Coprinellus micaceus*	Not specified	*Coryanebacterium xeroides*	Not specified	- Natural bioactive compounds extracted from *C. micaceus*	[99]
*Aureobasidium pullulans*	Tomato late blight Blue mold Gray mold Rhizopus rot Botrytis rot Penicillium rot Monalinia rot	*Phytophthora infestans* *Botrytis cinerea* *Penicillium expansum* *Penicillium* spp. *Monilinia laxa*	Tomato seeds Tomato Peach Apples Cherries Grapes Bananas	- Not specified	[21,58,100,101]
*Epicoccum nigram* *E. sorghinum*	Late blight Fusarium wilt Esca disease	*F. graminearum* *F. verticillioides* *F. oxysporum* *F. avenaceum,* *Colletotrichum falcatum* *Ceratocystis paradoxa* *Xanthomomas albilineans* *Pythium irregulare* *Phytophthora infestans* *Phaeomoniella chlamydospora* *Phaeoacremonium minimum*	Tomato Peas	- Natural bioactive compounds extracted from *Epicoccum* spp.	[3,102,103,104]
*Preussia africana*	Fusarium wilt Alternaria rot Blast disease	*F. solani* *C. albicans* *Pyricularia grisea*	Tomato seeds Rice	- Not specified	[105,106]

## 5. Mechanisms of Action of Fungal BCAs

Our current knowledge and understanding of the mechanisms by which fungal antagonists function is dependent on our ability to comprehend the interconnections among the pathogen, antagonist, and host tissue. These interconnections, known as tritrophic interactions (Figure 1), occur at the site of infection on the fruit and other fresh produce [36]. Nonetheless, these intricate and the precise mechanisms by which fungal BCAs affect microbial pathogens is not yet fully understood [1], largely due to the challenges associated with their studies, as their in vitro screening does not always translate into success in vivo). However, fungal BCAs can influence pathogens through various mechanisms, including, among others, the production of siderophores, secretion of metabolites, induction of host resistance, competition for nutrients and space, tolerance to high levels of reactive oxygen species, and direct parasitism [3].

### 5.1. Competition for Nutrients and Space

Competition for nutrients and space at the wound site between the fungal BCAs and the pathogen is the basic attribute in biocontrol and plays the most important role in diseases causing decay on fresh produce [1,3,30]. For more effectiveness, fungal BCAs should have the capacity to establish themselves at the wound site more quickly than the pathogen [1,3,41,106,107]. Thus, adaptation of fungal BCAs to several nutritional and environmental conditions should be superior to that of the pathogen. Researchers have reported positive outcomes in this aspect. For example, *P. guilliermondii* is effective against *P. digitatum* [79], and *Aureobasidium pullulans* against *P. expansum* [108] through competition of nutrients and space. In vitro investigations of these interactions indicate that through direct attachment, antagonistic fungal BCAs assimilate nutrients more swiftly than the target pathogens, thereby establishing themselves and limiting the germination of pathogen spores at the wound site [3]. Moreover, competition for resources occurs via iron depletion; however, this qualitative connection is significantly contingent upon proliferation and thriving of the fungal BCAs at the wound site [1,3].

### 5.2. Production of Metabolites

Fungal antagonists produce an array of antimicrobial metabolites such as antibiotics, lytic enzymes, and acids [75]. Among these metabolites, antibiotics are considered the second most significant mechanism by which fungal BCAs inhibit pathogens [3]. Antibiotic production has been reported for isolates of *Trichoderma* spp. [109]. *Trichoderma virens* strain P produces an antibiotic called gliovirin, effective against *P. ultimum* but not *Rhizoctonia solani.* Strain Q produces an antibiotic gliotoxin, highly effective against *R. solani* but less effective against *P. ultimum* [110]. Currently, researchers are prioritizing the development of non-antibiotic-generating fungal antagonists to manage postharvest infections of fresh produce. This shift is due to the harmful impact of antibiotics on both consumer health and the environment [79]. Given the current debate on antibiotic resistance of human pathogens, it is doubtful that an antibiotic-producing biocontrol agent would be registered for use on food or feed.

There are however other non-antibiotic antifungal compounds that have been reported. Examples include phytotoxins such as scopoletin and scoparone in citrus fruit [111]. Additionally, microbial antagonists in fruit wounds can produce substantial amounts of extracellular mucilage along the cell walls, which are involved cell adhesion. This mucilage comprises chemical compounds that actively induce a response or reaction, playing a role in signal recognition, identification, and subsequent reactions, thereby contributing to protective mechanisms [111]. Furthermore, formation of biofilm has also been reported by Settier-Ramírez et al. [58], where the formation of biofilm in liquid culture by the strain *S. cerevisiae* M25 was closely linked to its capacity to function as a biocontrol agent. The infection of *P. expansum* was effectively inhibited by *Candida sake,* which forms biofilms on the surface of apple fruit. The biofilm prevents the adhesion and growth of pathogenic fungi through competition for space and resources on the fruit surface. Moreover, oligosaccharide fragments derived from yeast cell wall polysaccharides are recognized as potent inducers of host defensive responses. Santhanam et al. [112] reported that the yeast *Pseudozyma flocculosa* produces extracellular fatty acids that control powdery mildew. Fungi are also involved in the production of lytic enzymes such as chitinase, glucanase, and proteinase, which aid in the degradation of the cell walls of the pathogen [3,24]. Grady et al. [113] demonstrated that *P. anomala* strain K yields high concentrations of ß-1,3-glucanase enzyme. Agirman et al. [114] reported on the stimulation of ethylene production in grape fruit by *P. guilliermondii* and *Aureobasidium pullulans*. *Candida saitoana* induces accumulation of ß-1,3-glucanases, chitinases, and peroxidases in apples [24,115,116]. Other examples are *P. membranaefaciens*, which antagonizes *Monilinia fructicola, Penicillium expansum, Rhizopus stolonifera*, and *Monilinia fructicola* on apples. Similarly, *C. albidus* antagonizes *P. expansum* on apples [114] while *P. guilermondii* [98] antagonizes *B. cinerea* [24].

In summary, BCAs frequently exhibit antimicrobial activities via the production of several metabolites, including (i) antibiotics, which hinder pathogen growth by interfering with the synthesis of cell walls, proteins, and nucleic acids [115]; (ii) volatile organic compounds (VOCs), which can impede pathogen development through several methods, such as membrane rupture or interference with signaling pathways [111,113,114]; (iii) exoenzymes, which have the ability to break down the cell walls of competing microorganisms, thereby strengthening the dominance of the BCAs in its environment; and (iv) indole and other molecules, which are signaling molecules that can influence the behavior of other bacteria, impacting their development and pathogenicity [117].

### 5.3. Siderophores Production

The other mechanism by which BCAs suppress plant diseases is through the synthesis of siderophores [118]. Siderophores are iron-chelating compounds with low molecular weight that are produced by fungi and bacteria under iron stress conditions [119]. Research has demonstrated that competition for iron is a significant factor in the suppression of postharvest diseases by antagonistic fungi. Siderophore production by fungal antagonists came into the spotlight when Klebba et al. [120] reported the production of an organoiron pigment by *Ustilago sphaerogena*, a rust fungus [120]. The yeast *Metschnikowia pulcherrima* produces a siderophore called pulcherrimin, which has been shown to impede the proliferation of certain postharvest fungal infections [121]. Likewise, the siderophore rhodotorulic acid, produced by *Rhodotorula glutinis*, effectively suppressed grey mold in apples [121]. Typically, microbial siderophores are classified into three major groups, namely, carboxylates, hydroxamates, and catecholates, based on the chemical properties of their coordination sites with iron [122,123]. Some siderophores are classified as phenolates [124], while others are categorized as mixed, containing both catecholate functional groups and hydroxamate [125]. Fungi produce siderophores in response to low iron availability. These high-affinity iron-chelating chemicals assist fungi in extracting iron, essential for their development and metabolism, from their surroundings. Iron performs various functions for the fungi, including, among others, the following: (i) Iron acquisition: Siderophores bind to ferric iron (Fe^3^⁺) and enhance its absorption by fungal cells. This iron is essential for several enzymatic activities, including those related to cellulose production. (ii) Promotion of cellulose biosynthesis: Siderophores may influence the control of cellulose synthesis. The signaling pathways initiated by iron absorption can activate metabolic processes that enhance cellulose production. (iii) Biofilm formation: Siderophores can facilitate biofilm development by the fungus. Biofilms create a protective matrix that improves nutrient absorption and can stimulate cellulose formation, leading to improved BCA yields [119]. That being said, pathogens also require iron as an essential nutrient for their growth. Therefore, efficient binding of the iron by high-affinity iron-chelating compounds produced by the fungal antagonists deprives the pathogens of this crucial nutrient, which slows or stops pathogen growth.

### 5.4. Host Resistance

Fruit host resistance can also be induced by microbial antagonists [126]. This mechanism of action has been made known through the development of high-throughput sequencing technologies and DNA microarrays. Reports have indicated alterations in DNA expression in both yeast and host tissues, contributing to a more comprehensive comprehension of the functioning of biocontrol systems. For example, in one study, *C. saitana* induced chitinase activity in the cell walls of the fruit host (apples) and established a structural barrier (papillae) against *P. expansum* [89]. In another study, microarrays were used to analyze the response of cherry tomatoes to the antagonistic yeast. A study on *C. laurentii* revealed an upregulation of genes associated with metabolism, signal transduction, and stress response. Conversely, genes involved in energy metabolism and photosynthesis were found to be downregulated. These alterations caused by *C. laurentii* resulted in heightened immunity against pathogenic infections [127]. Similarly, *A. pulilans* was shown to (i) cause a rise in 1,3-glucanase activity, as well as chitinase and peroxidase enzymes; (ii) stimulate the process of wound healing; and (iii) trigger defensive responses against *P. expansum* in apples [78].

Furthermore, application of the yeast *M. fructicola* on peach fruit induced upregulation of the genes involved in the pathogenic process and those that are part of the mitogen-activated protein kinase (MAPK) cascade, which is responsible for defensive signaling. On the other hand, the genes responsible for antioxidant activity, such as those coding for peroxidase, superoxide dismutase, and catalase, were downregulated [128]. The set of genes that were activated by *M. fructicola* in grape fruit showed a similar pattern with an induced resistance response. It was hypothesized that this induced response contributed to the effectiveness of *M. fructicola* in combating postharvest pathogens such as *P. digitatum* [128]. Supplementary to these findings, Zhao et al. [129] conducted a proteomic study and found out that *P. membranaefaciens* stimulated the production of antioxidant- and pathogenesis-related (PR) proteins in peaches. These proteins play an essential role in the control of *P. expansum* [129].

The MOA of fungal BCA development via host resistance entails the interactions between fungi and their host plants, which includes the following: (i) Host recognition: Fungi identify certain signals from the immune system of the host during invasion. This signal recognition can elicit defensive reactions in the host, including the synthesis of reactive oxygen species (ROS) and antibacterial substances. (ii) Fungal stress response: In reaction to host defense systems, fungi initiate their stress response pathways. This may result in heightened synthesis of secondary metabolites, such as siderophores, which facilitate the acquisition of vital nutrients like iron. (iii) Siderophore production: Fungi generate siderophores to acquire iron under stress, thus augmenting their capacity for cellulose synthesis, hence enhancing BCA growth. Iron is an essential cofactor for enzyme cellulose synthase. (iv) Interaction with host defense mechanisms: The presence of cellulose and other biopolymers enables the fungus to circumvent host defenses, offering a protective barrier against antimicrobial agents. This contact enables the fungus to endure and flourish within the host. (v) Regulatory networks: The fungal response to host resistance is governed by intricate signaling pathways. These pathways regulate the synthesis of siderophores, cellulolytic enzymes, and other elements that facilitate BCA formation [130]. These mechanisms are the interconnectedness between the MOA of the fungal antagonists against the pathogen. This is desirable since it is likely that problems similar to those experienced today, such as development of resistant pathogen strains, would probably occur if a one-substance effect was the only mechanism involved in pathogen inhibition.

#### Tolerance to High Levels of Reactive Oxygen Species (ROS)

Fungi frequently experience elevated concentrations of reactive oxygen species (ROS) as a result of environmental stressors. Antagonistic yeasts induce ROS signaling in host tissue, thereby acting as elicitors and activating host defenses [116,131,132,133]. This premise is supported by studies examining the expression of genes and proteins during yeast–fruit interactions [129,134]. Proteomic analysis revealed that *P. membranaefaciens*, an antagonistic yeast, induced production of six antioxidant proteins in peach fruit. These proteins included peroxiredoxin, glutathione peroxidase, and catalase. Additionally, biocontrol agents must be resistant to oxidative stress caused by reactive oxygen species (ROS), as this can compromise their viability and effectiveness [135]. Transcriptome analysis revealed that the application of *M. fructicola* induced the activation of peroxidases and superoxide dismutase in grapefruit [133]. Anwar et al. [136] examined the correlation between the ability to withstand oxidative stress and the overall fitness of postharvest biocontrol yeasts *Rhodotorula glutinis* LS-11 1 and *Cryptococcus laurentii* LS-28 [136]. *C. laurentii* LS-28 demonstrated superior biocontrol efficacy and elevated colonization of apple fruit in comparison to *R. glutinis* LS-11, owing to its greater resistance to ROS-induced oxidative stress. Verma et al. [109] investigated the oxidative stress responses of *Cystofilobasidium infirmominiatum* PL1, *M. fructicola*, and *C. oleophila* [110,137,138]. Their results indicated highest tolerance to exogenous H_2_O_2_ by *M. fructicola,* whereas *C. infirmominiatum* exhibited the highest sensitivity [110,137,138]. Specifically, after a 20 min incubation, the survival rate for *M. fructicola* was 88% in 200 mM H_2_O_2_, 28% in 100 mM H_2_O_2_ for *C. oleophila*, and 23% in 20 mM H_2_O_2_ for *C. infirmominiatum*. Li et al. [139] reported that a substantial drop in the viability of *Pichia caribbica* as the concentration of H_2_O_2_ increased from 5–20 mM. The majority of the yeast cells perished after sixty minutes of exposure to 20 mM H_2_O_2_. In addition to oxidative stress, antagonistic yeasts might be confronted with reduced oxygen levels that accompany storage in a controlled environment [136,140].

Tolerance of the antagonistic fungi experience to high levels of ROS generally operates as follows: (i) Synthesis of antioxidants: Fungi generate a range of antioxidant substances, including superoxide dismutase (SOD), catalase, and glutathione. These antioxidants neutralize reactive oxygen species, safeguarding cellular integrity and facilitating regular metabolic activities. (ii) Changes in gene expression: The oxidative stress response can influence the expression of genes, such as upregulation of genes involved in cellulose production. This enables the fungus to allocate resources for cellulose biosynthesis, which preserves cellular health, thereby enhancing BCA synthesis. (iii) Cell wall remodeling: ROS activate signaling pathways that induce changes in the composition of the fungal cell wall, hence improving the structural integrity of the fungus. This remodeling can enhance cellulose synthesis. (iv) Metabolic alterations: Exposure to ROS can induce changes in fungal metabolism, prioritizing pathways that enhance cellulose production and other stress responses. This versatility enables fungi to flourish in adverse conditions [133,134,137,138].

### 5.5. Direct Parasitism

Direct parasitism occurs when a particular microbial antagonist directly attacks and kills the pathogen and/or its propagules [24]. This phenomenon is often known as mycoparasitism. Parasitism is dependent on intimate recognition and interaction between the pathogen and the antagonist, secretion of lytic enzymes, and the parasite’s vigorous proliferation within the host [79,88]. Studies on the direct parasitic role of fungal BCAs in suppression of postharvest fruit diseases are limited [137]. Masumoto et al. [140] demonstrated that *P. guilliermondii* cells can attach to *P. italicum* hyphae. Likewise, antagonistic yeast *C. saitoana* has been reported to directly parasitize *P. italicum* [141]. Microbial antagonists display firm adherence properties. Considering this, along with heightened effectiveness of cell wall-disintegrating enzymes that facilitate penetration by mycoparasites, it is likely that antagonists have a substantial impact on biocontrol activity.

The mechanism of action through direct parasitism entails the following: (i) Host invasion: Parasitic fungi infiltrate host tissues employing specialized structures, like haustoria or appressoria, to penetrate and assimilate resources. This invasion frequently elicits a defensive reaction from the host. (ii) Nutrient acquisition: By parasitizing their host, fungi extract vital resources such as carbohydrates and proteins. The acquisition of this nutrient is essential for fungal growth and metabolism, facilitating cellulose synthesis. (iii) Defense evasion: The synthesis of cellulose and other polysaccharides can provide a protective barrier around fungal cells, safeguarding them from host defenses. This modification enables the fungus to endure and flourish within the host. (iv) Symbiotic interactions: In certain instances, the parasitic fungi may participate in intricate relationships with their hosts, through which both species obtain mutual benefits. This may enhance cellulose synthesis when the fungus acclimatizes to its ecological environment [88,140,141,142,143,144].

There are, however, other novel discoveries of the mechanism by which antagonistic yeasts exert their activity. These include a reduction in fruit oxidative injury [141], their capacity to tolerate salt [142], adjustments to population density, and morphological changes [142]. More research exploring these mechanisms of action is needed, as there is currently limited information on them. A deeper understanding of these mechanisms of action will reveal innovative strategies for proper and effective use of fungal antagonists as biocontrol agents.

## 6. Challenges and Difficulties in Establishing Beneficial Microbes as BCAs

Despite being well documented, BCAs currently hold a commercial value of less than 5% of the entire fresh produce protection industry [145]. The limited number of licensed products for biocontrol of plant diseases is significantly linked to the low technology transfer, implying that the agricultural sector, mainly in developing countries, has yet to recognize their economic potential. Besides that, for promising microbial agent candidates, there is a lack of sufficient knowledge regarding the organism-specific research methods for large-scale production and development, making mass production of the entire microbe in in vivo conditions both expensive and time-consuming [108].

Notwithstanding the potential advantages of BACs, there are several obstacles to overcome in their application, particularly when transitioning from laboratory research to large-scale field use [47]. Selecting an effective BAC strain requires rigorous laboratory screening processes based on well-established mechanisms such as the activity of antimicrobial compounds and extracellular enzymes [146]. However, laboratory screening does not always translate into success under field conditions, as isolates demonstrating strong biocontrol capabilities in vitro may not perform equally well in more complex, real-world environments [147].

Ensuring BCAs reach the right area at the right time, in sufficient density to be effective, and remain there permanently, are some of the most challenging aspects of their use. Since biocontrol involves the introduction of non-native living organisms, their application may come with associated serious ecological impacts. Non-native species, for instance, may become invasive and trigger harmful environmental consequences if they spread beyond the area where they were introduced [145]. Moreover, some BCAs exhibit high efficacy under in vitro laboratory conditions, but ecological restrictions under real full-scale conditions may hamper their performance [145,148], making them economically non-viable and less appealing to users. Most BCAs have a limited shelf life, and practical challenges such as maintaining viability during storage and transport, applying the agents effectively, and ensuring consistent results can hinder their widespread adoption [149]. Effective application technologies for BCAs may not be readily available or accessible. Biocontrol agents require specialized storage and handling procedures to maintain viability and efficacy. Lack of clear regulatory guidelines and standards for BCAs can create uncertainty and barriers to their use [150]. Despite efforts to identify effective antagonists, several promising microorganisms have not advanced to the next stage of research, primarily due to challenges associated with formulation trials and manufacturing scaling up [147].

BCAs, like synthetic pesticides, are submitted to risk assessments regulated by European Regulation (EC) No. 1107/2009 before being approved for commercialization. This regulation prevents the approval of substances that present unacceptable risks to human/animal health and the environment [35]. By addressing these challenges and limitations, BCAs can become more mainstream and widely adopted, providing a more sustainable and environmentally friendly alternative to conventional control methods.

## 7. Conclusions and Future Trends

There are several strategies, ranging from physical, chemical, and recently, biological practices, which are currently used for the control and management of fresh produce postharvest decay. Although some of these approaches, e.g., chemical treatments, are considered the primary methods for control of pathogens causing the postharvest diseases of fresh produce, they all have several limitations that hinder their continued use. Physical methods are undesirable, as their application can damage fresh produce, thereby further rendering it more susceptible to infections. They are also relatively expensive and thus may not be affordable for smallholder farmers. There are serious public concerns over the use of chemical treatments due to their ecotoxicity and human toxicity, contribution to development of antibiotic resistance in pathogens, and their associated exorbitant costs. Furthermore, none of the currently used methods have been able to control these diseases completely. For these reasons, continuous development of more natural, environmentally friendly, and affordable methods is of critical importance.

Biological control, using microorganisms as biological control agents (BCAs), holds promise as such an alternative. These microorganisms are preferred because they are safer, cheaper, naturally available, have minimal side effects, and are acceptable world-wide. Fungal antagonists are particularly suitable due to the following properties: (i) they grow rapidly on surfaces of fresh produce; (ii) they are effective at low dosages; (iii) as opposed to traditional chemicals, they reduce the risk of development of resistance in pathogens, as they operate through multiple mechanisms of action; (iv) they tend to be more stable than chemical fungicides when stored properly; (v) they have lower re-entry intervals; (v) they are less phytotoxic; and (vi) are accessible to small-scale producers.

The use of fungal antagonists for biological control will not only reduce postharvest fruit losses, but will also preserve the nutrient content of the harvested commodities as compared to the use of fungicides and will potentially also improve smallholder farmers’ income. We therefore recommend that more studies be conducted on the use of antagonistic fungi in the control and management of fresh produce postharvest. Continuous advancements in formulation, isolation, and application methods, particularly after harvest (i.e., during storage), can increase the efficacy of fungal antagonists as BCAs. While challenges such as inconsistent performance, cost barriers, and regulatory complexities persist, ongoing research and innovations are paving the way for more effective applications of BCAs. Along with the previously mentioned factors, the success of biological control also depends on the ability of scientists to comprehensively document and analyze research outcomes. This practice allows for researchers to identify weaknesses in BCAs, such as lack of efficacy, uneven field performance, or unfavorable economic considerations, enabling them to address these issues in future studies.

## Figures and Tables

**Figure 1 jof-11-00082-f001:**
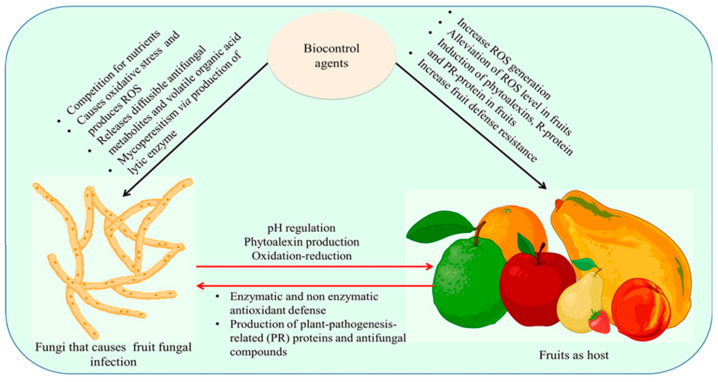
The relationship between biocontrol, pathogen, and host (mechanism of biocontrol action) is an intricate process consisting of multiple developments. The biocontrol agents impact fungal pathogens by instigating oxidative stress that triggers ROS, releases volatile organic acid and diffusible antifungal metabolites, and mycoparasitism via the production of lytic enzymes. In fruits, induction of phytoalexins R-protein and PR-protein and an upsurge in ROS generation lead to an increase in fruit defense resistance by antioxidant enzymes.

## Data Availability

Not applicable.

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
