# Peer review of "Fungal Biocontrol Agents in the Management of Postharvest Losses of Fresh Produce—A Comprehensive Review"

_jof, 2025, doi:10.3390/jof11010082_

Round 1
Reviewer 1 Report (New Reviewer)
Post-harvest decay of vegetables and fruits presents is a big problem that not only leads to huge economic loss but also cause mycotoxin contamination which pose a potential threat to human health. This review summarized the methods currently used for management of post-harvest diseases of fresh produce, highlighting their limitations,
and the use of biocontrol agents as an alternative strategy for control of such diseases, with emphasis on fungal antagonists, their mode of action, and more importantly, their advantages when compared to other methods commonly used. However, in order to improve the quality of the review, some comments should be addressed.
General comments
1. For “Characteristics of ideal fungal BCAs”, how about the practical application of the ideal fungal BCAs.
2. For “3.3 Delivery systems of fungal BCAs”, in fact, there are two treatment mode, one is pre-harvest application, and another one is post-harvest application. I think there are no delivery system.
3. Line 343-346, about “biofilm” how to control the disease of produce, and how about the main component.
4. Figure 1 is a little simple, please detail the mode of action.
5. Please explain the “Siderophores production”, and how the Siderophores to participate and control the disease of produce.
6. 5.4 Host resistance and 5.5 Tolerance to high levels of reactive oxygen species (ROS), in fact, they are attribute to induce the host resistance, because low concentration ROS can be act as signal molecular to activate the downstream signal pathway and up-regulate the enzymatic activity such as SOD, POD, CAT etc.
7. The English writing of the whole manuscript needs to be improved.
Post-harvest decay of vegetables and fruits presents is a big problem that not only leads to huge economic loss but also cause mycotoxin contamination which pose a potential threat to human health. This review summarized the methods currently used for management of post-harvest diseases of fresh produce, highlighting their limitations,
and the use of biocontrol agents as an alternative strategy for control of such diseases, with emphasis on fungal antagonists, their mode of action, and more importantly, their advantages when compared to other methods commonly used. However, in order to improve the quality of the review, some comments should be addressed.
General comments
1. For “Characteristics of ideal fungal BCAs”, how about the practical application of the ideal fungal BCAs.
2. For “3.3 Delivery systems of fungal BCAs”, in fact, there are two treatment mode, one is pre-harvest application, and another one is post-harvest application. I think there are no delivery system.
3. Line 343-346, about “biofilm” how to control the disease of produce, and how about the main component.
4. Figure 1 is a little simple, please detail the mode of action.
5. Please explain the “Siderophores production”, and how the Siderophores to participate and control the disease of produce.
6. 5.4 Host resistance and 5.5 Tolerance to high levels of reactive oxygen species (ROS), in fact, they are attribute to induce the host resistance, because low concentration ROS can be act as signal molecular to activate the downstream signal pathway and up-regulate the enzymatic activity such as SOD, POD, CAT etc.
7. The English writing of the whole manuscript needs to be improved.
Author Response
Thank you so much for taking your time to go through our manuscript, may you kindly find attached point by point response to your comments.
Kind regards

Reviewer 2 Report (New Reviewer)
Ramudingana and the other authors have created a very well written and detailed review of Fungal BCAs. This primer is very successful in covering all the key areas/aspects related to the research and use of Fungal BCAs specifically used in produce. I would of liked to see an expansion of the section 2 titled "Characteristics of ideal fungal BCASs". It would be very helpful to go over a few of the current BCAs in use and explain how they posses all or most of the traits that you have listed. The authors mention multiple species in the tables present but do not go into much detail about any particular fungal BCA in the text.
Page numbers starting on Page 11 of the manuscript reset and start numbering at 1.
Line 157 and 169, this is the first mention of Canidia sake and Pichia anomala but the abbreviation of theirs name are used (i.e. C. sake). Please correct this.
Page 10 in Table 2, for A. pullulans, the column for "Mechanism of activity" should be changed to "-Not specified"
Line 345 I fairly certain "S. serevisiae" should be "S. cerevisiae".
Author Response
Thank you so much for taking your time to go through our manuscript, may you kindly find attached point by point response to your comments.
Kind regards

Reviewer 3 Report (New Reviewer)
I would like to highlight 3 things:
(1) Biocontrol agents have been around for many decades, yet they have never been mainstream control methods, despite their benefits detailed in this manuscript. This review would be strengthened by addressing the challenges and limitations of biocontrol agents, including their lower efficacy compared to conventional measures, practical difficulties in implementation, and other issues that prevent widespread adoption in real-world situations.
(2) The efficacy of fungal biocontrol agents is often evaluated in vitro, which may not represent their performance in real-world setting. The authors should clarify whether the research cited in the manuscript was conducted in laboratory settings or real commercial setting, and which products have been validated their efficacy in real-world setting?
(3) Terms like "significantly reduced" or "effectively controlled" are ambiguous without economic context. The authors should address whether the biocontrol products outperformed untreated controls and if the level of decay control is economically acceptable.
100% untreated fruit got decay - vs. - 80% fruit treated with the product X got decay. Although the product X "significantly reduced" fruit decay, there is almost no practical value for the product X due to its poor performance.
Some minor comments are in the attached pdf file.

Author Response
Thank you so much for taking your time to go through our manuscript, may you kindly find attached point by point response to your comments.
Kind regards

Round 2
Reviewer 1 Report (New Reviewer)
The author responsed all the comments, now it can be accept.
The author responsed all the comments, now it can be accepted.
Author Response
Dear Reviewer,
thank you so much for accepting our manuscript based on response we made on your comments, however we are uploading the same response again as the system says we have not reply.
we are doing so so that the minor comments from the reviewer 3 can be accepted by your system.
Kind regards,
Tshifhiwa

Reviewer 3 Report (New Reviewer)
The authors revised the manuscript and incorporated reviewers' comments and suggestions. I recommend the publication of this manuscript after the authors address the further editorial comments and suggestions.
L112: (vi) do not leave any residues. They do leave some residue, including the BAC itself, buffer or other reagents in the commercial fungicide. But the important aspect of the BAC is that such residue is often exempt from MRL (Maximum Residue Limits).
L292: The development of antagonists from filamentous fungi as biocontrol agents for managing postharvest diseases is less advanced compared to yeasts.
Remove "mold" and "when".
L569-L581: Shorten this part in a sentence. Focus on "challenges and difficulties", we don't need to see all the details of such regulation. Just cite regulation article.
L617-L620: The success of biological control also depends on scientists' ability to comprehensively document and analyze research outcomes. This practice allows researchers to identify weaknesses in BCAs, such as lack of efficacy, uneven field performance, or unfavorable economic considerations, enabling them to address these issues in future studies.
Remove "negative data"
Reference #9 (Wen et al. 2024) is cited too often for general ideas in the manuscript, but this paper #9 is just a research paper. All those general ideas are not scientifically demonstrated in this research paper #9. Use other sources, book, review articles for general ideas.
Author Response
Thank you so much for taking your time to go through our manuscript, may you kindly find attached point by point response to your comments.
Kind regards

This manuscript is a resubmission of an earlier submission. The following is a list of the peer review reports and author responses from that submission.
Round 1
Reviewer 1 Report
Fungi are part of a group of organisms with great phylogenetic diversity and ecological and economic relevance. Entomopathogenic fungi often penetrate the bodies of insects, forming spores that, when spread (through wind, rain or in the presence of other contaminated insects), cause potential epizootics.
In some countries, the production of entomopathogenic fungi is normally carried out based on active products from mycoses, to control insect pests, and is one of the most effective alternatives to be used in the biological control of pests or insect vectors. In this context, the use of entomopathogenic fungi in biological control stands out for not causing great damage to the environment, generating better quality in crops or wherever they are applied, due to the reduction in the use of chemical products.
The problem of loss of vegetables and fruits before harvest is a major problem for sustainable food production worldwide and, in recent times, the application of synthetic fungicides has become the most popular technique for managing post-harvest losses. However, there are concerns and reported evidence of hazardous impacts on consumer health and the environment traceable to the application of fungicides as preservatives in fresh produce. Therefore, healthier and environmentally friendly alternatives to synthetic chemicals in managing postharvest spoilage of fresh produce should be advocated. There is a growing consensus that the use of biological control agents (BCAs), mainly fungi, is more sustainable and effective in controlling postharvest losses compared to chemical pesticides. Secretion of antifungal compounds, parasitism, and competition for nutrients and space are the most commonly used mechanisms of antagonistic actions of these BCAs. This review article submitted by Phathutshedzo Ramudingana et al., provides an overview of the use of fungal antagonists as agents for controlling postharvest spoilage and also provides information on the currently used fungal antagonists and highlights the mode of action of these antagonists. Therefore, we hypothesize that the use of fungal antagonists in preventing post-harvest losses of fresh produce is more effective when compared to the use of chemicals.
The review is interesting and extremely relevant to the field of study
Fungi are part of a group of organisms with great phylogenetic diversity and ecological and economic relevance. Entomopathogenic fungi often penetrate the bodies of insects, forming spores that, when spread (through wind, rain or in the presence of other contaminated insects), cause potential epizootics.
In some countries, the production of entomopathogenic fungi is normally carried out based on active products from mycoses, to control insect pests, and is one of the most effective alternatives to be used in the biological control of pests or insect vectors. In this context, the use of entomopathogenic fungi in biological control stands out for not causing great damage to the environment, generating better quality in crops or wherever they are applied, due to the reduction in the use of chemical products.
The problem of loss of vegetables and fruits before harvest is a major problem for sustainable food production worldwide and, in recent times, the application of synthetic fungicides has become the most popular technique for managing post-harvest losses. However, there are concerns and reported evidence of hazardous impacts on consumer health and the environment traceable to the application of fungicides as preservatives in fresh produce. Therefore, healthier and environmentally friendly alternatives to synthetic chemicals in managing postharvest spoilage of fresh produce should be advocated. There is a growing consensus that the use of biological control agents (BCAs), mainly fungi, is more sustainable and effective in controlling postharvest losses compared to chemical pesticides. Secretion of antifungal compounds, parasitism, and competition for nutrients and space are the most commonly used mechanisms of antagonistic actions of these BCAs. This review article submitted by Phathutshedzo Ramudingana et al., provides an overview of the use of fungal antagonists as agents for controlling postharvest spoilage and also provides information on the currently used fungal antagonists and highlights the mode of action of these antagonists. Therefore, we hypothesize that the use of fungal antagonists in preventing post-harvest losses of fresh produce is more effective when compared to the use of chemicals.
The review is interesting and extremely relevant to the field of study
Reviewer 2 Report
1. In the abstract and conclusion, a comparison of existing antimicrobial agents should be presented.
2. There has been a decrease in relevant research articles in recent years.
3. When introducing physical, chemical, and biological control methods, a more in-depth and comprehensive analysis of the advantages and disadvantages of each method can be conducted, especially emphasizing the advantages of biological control methods.
4. Although various mechanisms such as competition for nutrients and space, production of metabolic products, siderophores, induction of host resistance, tolerance to reactive oxygen species, and direct parasitism have been introduced, the discussion of the specific molecular biological processes and related signaling pathways for each mechanism is not deep enough.
5. The conclusion merely emphasizes the potential and importance of yeast fungi as biological control agents, which is not quite appropriate since the article's theme is to summarize a variety of biological control agents.
1. In the abstract and conclusion, a comparison of existing antimicrobial agents should be presented.
2. There has been a decrease in relevant research articles in recent years.
3. When introducing physical, chemical, and biological control methods, a more in-depth and comprehensive analysis of the advantages and disadvantages of each method can be conducted, especially emphasizing the advantages of biological control methods.
4. Although various mechanisms such as competition for nutrients and space, production of metabolic products, siderophores, induction of host resistance, tolerance to reactive oxygen species, and direct parasitism have been introduced, the discussion of the specific molecular biological processes and related signaling pathways for each mechanism is not deep enough.
5. The conclusion merely emphasizes the potential and importance of yeast fungi as biological control agents, which is not quite appropriate since the article's theme is to summarize a variety of biological control agents.